# Effects of Indole-3-Lactic Acid, a Metabolite of Tryptophan, on IL-4 and IL-13-Induced Human Skin-Equivalent Atopic Dermatitis Models

**DOI:** 10.3390/ijms232113520

**Published:** 2022-11-04

**Authors:** Kyunghee Kim, Hyeju Kim, Gun Yong Sung

**Affiliations:** 1Interdisciplinary Program of Nano-Medical Device Engineering, Hallym University, Chuncheon 24252, Korea; 2Integrative Materials Research Institute, Hallym University, Chuncheon 24252, Korea; 3Major in Materials Science and Engineering, Hallym University, Chuncheon 24252, Korea

**Keywords:** indole-3-lactic acid, atopic dermatitis, skin equivalent, tryptophan metabolism, Th2 cytokine, interleukin-4, interleukin-13

## Abstract

Indole-3-lactic acid (I3LA) is a well-known metabolite involved in tryptophan metabolism. Indole derivatives are involved in the differentiation of immune cells and the synthesis of cytokines via the aryl hydrocarbon receptors for modulating immunity, and the indole derivatives may be involved in allergic responses. I3LA was selected as a candidate substance for the treatment of atopic dermatitis (AD), and its inhibitory effect on AD progression was investigated. Full-thickness human skin equivalents (HSEs) consisting of human-derived cells were generated on microfluidic chips and stimulated with major AD-inducing factors. The induced AD-HSEs were treated with I3LA for 7 days, and this affected the AD-associated genetic biomarkers and increased the expression of the major constituent proteins of the skin barrier. After the treatment for 14 days, the surface became rough and sloughed off, and there was no significant difference between the increased AD-related mRNA expression and the skin barrier protein expression. Therefore, the short-term use of I3LA for approximately one week is considered to be effective in suppressing AD.

## 1. Introduction

Atopic dermatitis (AD) is a common chronic inflammatory skin disease [1]. The pathological features of AD include dry erythematous lesions and severe itching, extensive intercellular edema (spongiosis) in the epidermis, and intracellular edema (i.e., ballooning of the keratinocytes) [2,3]. In AD, T cells are hyperdifferentiated into T helper 2 (Th2) cells, and the mRNA expression levels of the Th2 cytokines such as interleukin (IL)-4, IL-5, IL-13, and interferon-γ (IFN-γ) are significantly increased when they are compared to those in normal skin [3]. AD is a Th2-mediated hyperimmune response, and approximately 80% of AD patients have elevated serum IgE levels. The Th2 cytokines IL-4 and IL-13 are significantly overexpressed in the skin lesions of patients with AD. The weakening of the epidermal barrier function in AD is associated with the downregulation of barrier-associated molecules such as filaggrin (FLG), loricrin (LOR), and involucrin (IVL) [4], which are coordinated by various barrier proteins such as FLG and LOR via sequential cross-linking. Most human epidermal barrier proteins are mapped to the epidermal differentiation complex (EDC) by a 1.9 Mbp gene cluster that is located on chromosome 1p21, which plays a substantial role in the terminal differentiation of the human epidermis [5]. A variety of external and internal stimuli, including Th2 (IL-4 and IL-13) or Th22 (IL-22) cytokines, lead to the upregulation or downregulation of the barrier gene expression levels of the barrier proteins in EDCs. IL-4 and IL-13 are known to significantly inhibit the gene and protein expression of FLG and LOR [6,7,8]. IL-4 and IL-13 induce AD in skin keratinocytes [9]. Thymic stromal lymphopoietin (TSLP) is a major epithelia-derived cytokine that is involved in AD pathogenesis [10,11]. In particular, increased TSLP expression was observed in skin keratinocytes from AD patients, and elevated TSLP levels were reported in the serum of children with AD [12]. Corneocyte TSLP expression levels have been reported to be correlated with cutaneous sensitivity and asthma severity [13].

Because of its genetically determined pathophysiology, there is no perfect cure for AD. Until recently, AD treatments have focused on preventing the triggers, hydrating the skin, and reducing the inflammation. A typical treatment for atopy is the use of topical corticosteroids (steroids) to treat the inflammation. However, since the long-term use of topical corticosteroids is limited by its side effects [14,15,16,17], substances and treatments that can be used for a long time without the occurrence of toxicity are required. Treatment with microbial metabolites can be considered as a safer and more effective strategy for the treatment of AD. AD treatment strategies can consider modulating the skin microbiota to promote the development of topical or systemic drugs, which originate from microbial metabolites and the production of specific useful metabolites. Microbial metabolites have received widespread interest over the past few years, however, skin microbial metabolites are less well-known than gut microbes are [18,19]. AD is the most well-studied skin disease in microbiological treatments, and the research has mainly focused on the structure and diversity of the microbial communities [20,21]. Previous studies have revealed significant changes in the skin’s microbial community diversity and structure in AD patients compared to healthy individuals as well as changes in specific microbes, such as *Staphylococcus aureus* [21,22]. It has also been reported that tryptophan (Trp) metabolism is attenuated in the skin microbiome of patients with AD [23]. These results have identified major metabolites that are formed by the skin microbiota that are either current or likely targets in the search for AD therapeutics. Indole and indole derivatives are essential amino acid Trp metabolites that influence the host’s physiology via several molecular mechanisms [24,25]. Indole-3-lactic acid (I3LA) is a metabolite of Trp, and indole derivatives participate in the differentiation of immune cells and the synthesis of cytokines via the aryl hydrocarbon receptors (AhR) to regulate the immune responses and participate in anti-inflammatory and allergic reactions [26,27]. AhR is involved in skin homeostasis in both physiological and pathological conditions [28]. Recently, it was reported that microbial metabolites on the skin’s surface can activate AhR in keratinocytes and control skin inflammation [29]. Histamines act through the H4 receptor of T cells to inhibit the signal transducer and activator of transcription 1 (STAT1) activation and promote Th2 reactions to cause atopy [30]. I3LA was effective in regulating the STAT1 signaling pathway [31]. In a previous study, the macrophages that were differentiated from THP-1 cell lines were stimulated with lipopolysaccharide (LPS) to confirm the effect of an indole derivative on the expression of IL-4, IL-6, and IL-13, which validated that indole-3-butyric acid (IBA) and I3LA play a role in allergic pathogenesis [26].

Most models and disease assessments that predict drug toxicity at nonclinical stages have been conducted via animal testing. However, their biological similarities that they have with humans are often limited, and the strengthened regulations on the use of laboratory animals have led to a growing need for alternative experimental models [32,33,34]. In this study, AD was induced using a skin-on-a-chip generation of HSEs, which is an alternative animal model. In the HSEs composed of human-derived AD-induced cells, we attempted to confirm the effects of I3LA on AD disease progression and symptom alleviation, as well as the expression of related cytokine mRNAs, TSLP mRNA, and epidermal barrier proteins.

## 2. Results and Discussion

### 2.1. MTT Assay for Cytotoxicity Test

A concentration range which was not toxic to the cells was identified by applying varying concentrations of I3LA to the AD-HSEs. Using an MTT assay, the cell viability with respect to I3LA treatment at varying concentrations from 78.125 μM to 10 mM was calculated (Control = no I3LA) as shown in Figure 1. It was assumed that the concentrations that resulted in viabilities that were higher than 50% were safe [35], and among the concentrations at which the viability was 50% or higher, the I3LA concentration value, 1.25 mM, was selected.

### 2.2. Inhibitory Effect of Indole-3-Lactic Acid on Contraction

Contractions occur when the cultures are organized in a 3D manner, which is a challenge that still lacks a solution. This phenomenon can be an indicator of cell activity and culture conditions. Overall, the daily variability of the contraction (contraction percentage per day) tended to decrease as the air exposure period increased. This implies that once the HSEs reach a certain level of contraction, a further contraction does not occur. As shown in Figure 2b–d, the contraction was markedly suppressed by I3LA in the control samples without the cytokine addition. As shown in Figure 2, the changes in the contraction rate due to I3LA treatment were compared between the groups that were treated with IL-4 and IL-13, and the group that was treated with only IL-13. The AD-HSEs treated with IL-4 alone showed no difference in the contraction from that in the I3LA-treated groups. The AD-HSEs co-treated with IL-13 and I3LA showed a slight reduction in the contraction. The AD-HSEs co-treated with IL-4/IL-13 and I3LA also showed a decreased contraction. summarizes the quantitative changes in the contraction rate with and without the I3LA addition. When we were comparing the differences in contraction rates (Table 1), the treatment with I3LA suppressed the contraction by up to 71.27% in comparison to that in the control. In contrast, the treatments with IL-4 and I3LA had no effect on the contraction compared to that of I3LA alone. AD is caused by the excessive release of inflammatory cytokines such as IL-4 and IL-13 as the Th2 cells differentiate. IL-4 is mainly expressed in the late chronic stages of AD. The treatments with IL-4 and IL-13 decreased the FLG, LOR, and IVL expressions. The treatments with IL-4 and IL-13 alone reduced the FLG2 and hornerin (HRNR) gene expression. HRNR is structurally related to FLGs, and it is involved in epidermal barrier formation [36]. Overall, the inhibitory effect of I3LA on the contraction was reduced by the addition of cytokines, and the magnitude of the reduction was in the following order (from strongest to weakest): the group that was stimulated by IL-4 and IL-13, the group that was treated only with IL-13, and the group that was treated only with IL-4.

### 2.3. Inhibitory Effect of Indole-3-Lactic Acid on Contraction

The gene expressions were compared between the IL-4 and IL-13 treated groups between 7 and 14 days of the I3LA treatment. Experimental groups that were treated with IL-4 and IL-13 separately or together showed a decreased expression of FLG, LOR, and IVL, and an increased expression of carbonic anhydrase 2 (CA2) and TSLP. The IL-6 expression was increased in the AD-HSEs that were induced with cytokines after 7 days. This indicated that the IL-4 and IL-13 stimulation induced an AD-like response. The changes in the gene expression were compared after the I3LA treatment. The results in Figure 3a–c show that the treatment with I3LA for 7 days tended to restore the TSLP and IL-6 gene expression to the control levels and decrease the expression of both IL-4 and IL-13. In contrast, FLG, LOR, and IVL were downregulated or not affected by I3LA. CA2 [9], a gene that is known to be overexpressed in AD patients, showed a decreased expression after the combined IL-4/IL-13 treatment. Our results of the reduced TSLP expression and the alleviation of AD after 7 days of the treatment are consistent with the previous reports of locally applied inole-derivatives which partially alleviated the skin inflammation and inhibited the TSLP formation in keratinocytes [29]. In previous reports, when macrophages that are differentiated from THP-1 cell lines were stimulated with LPS, IBA significantly inhibited the LPS-induced upregulation of IL-4 and IL-6 mRNA, and I3LA inhibited the expression of IL-6 mRNA [26]. Our results also showed that I3LA suppresses the IL-6 mRNA expression. This indicates that I3LA regulates the expression of several cytokine mRNAs and may be involved in allergy-related mechanisms.

As shown in Figure 3d–f, the treatment with I3LA for 14 days appeared to increase the expression of CA2 and TSLP and decrease the expression of IL-6. When FLG was stimulated with IL-4, its expression was restored to that of the pre-I3LA levels. However, for IVL and LOR, the results only showed the upregulation of LOR after the cytokine stimulation. In conclusion, having upregulated TSLP, IL-6, and CA2 levels, as confirmed by the AD biomarkers, improved the symptoms of AD when it was treated with I3LA for 7 days. The treatment with I3LA for 14 days tended to worsen the AD. I3LA appeared to have no effect on the expression of the epidermal barrier-associated genes.

### 2.4. Skin Barrier Recovery Effect by I3LA

To confirm the effects of I3LA on the morphology, the tissue cross-sections were examined using hematoxylin and eosin (H&E) staining. As shown in Figure 4, when we were looking at the stratum corneum of the skin tissue on day 7, it was found that the stratum corneum in the group that was treated with I3LA was thicker than that of the untreated group. The immunohistochemistry (IHC) results confirmed that the treatment with I3LA for 7 days resulted in a thicker stratum corneum due to the increased expression of the key constituent proteins of the skin barrier, which led to the skin barrier’s restoration. On the 14th day, as shown in b, it was observed that the keratin layer of the group that was treated with I3LA was not significantly different or damaged when it was compared to the untreated group. The IHC results showed no significant differences in the skin barrier proteins after 14 days of the I3LA treatment.

The treatment with I3LA for 7 days increased the expression of the major constituent proteins of the skin barrier. The treatment for 14 days did not generate a significantly different expression of these proteins compared to the 7-day treatment. Similar to the results of the gene expression comparison using a PCR, I3LA did not affect the expression of epidermal barrier-related genes, and therefore, there was no significant difference in the expression of the skin barrier proteins.

IL-4 and IL-13 decrease the expression of FLG, IVL, and LOR, the major constituent proteins of the skin barrier [4,37,38,39]. To investigate the expression of FLG, IVL, and LOR in the AD tissue and the efficacy of I3LA in restoring the skin barrier, an IHC was performed, and the results are shown in Figure 4c–h. The response to the I3LA treatment was compared between the groups that were treated with IL-4 or IL-13, individually, and the group that was treated with both. When I3LA was administered for 7 days, FLG, IVL, and LOR were increased in the group that was not treated with I3LA. However, FLG and IVL decreased in the group that was treated with both IL-4/IL-13 and I3LA in comparison to the group that was not treated with I3LA. When I3LA was administered for 14 days, FLG increased in the group that was treated with IL-13 in comparison to the group not treated with I3LA. FLG expression decreased in the group that was treated with IL-4 in I3LA, and the skin barrier’s damage was not reversed. The expression of IVL and LOR was higher in the group that was treated with I3LA than it was in the untreated group. The total protein expression in the control group that was treated with I3LA for 14 days was reduced when it was compared to that of the group that was not treated with I3LA. Overall, the I3LA treatment for 7 days increased the expression of the major constituent proteins of the skin barrier, but the treatment for 14 days did not lead to further changes.

In order to morphologically confirm the effect of I3LA, the surface of the tissue was observed using SEM, and the results of this are summarized in Figure 5. Corneocytes were extruded into a hexagonal shape and sloughed off in the SC layer. Decreased FLG expression can affect the natural moisturizing factor (NMF) levels, and consequently, alter the skin’s hydration and pH [40]. Skin dehydration causes xerosis, the weakening of the epidermal barrier, and itchiness. An increased pH enhances the activity of the proteases that are responsible for keratin exfoliation and decreases the activity of the enzymes that are involved in barrier lipid synthesis [41,42]. The treatment with cytokines likely caused a severe desquamation of the stratum corneum and led to a rough surface that is indicative of hyperkeratosis. However, when the combined IL-4/IL-13 treatment was used for 14 days alongside I3LA, we observed the appearance of a very rough surface, the severe shedding of keratin, and/or cracked skin surfaces, suggesting that the barrier formation was severely compromised, and the AD had progressed. When we were examining the surface of the tissue, the AD-HSEs that were treated with I3LA for 7 days were observed to have no or reduced amounts of raised areas and a flat or smooth surface. This is thought to be related to the decrease in the expression of the TSLP, CA2, and IL-6 genes when it was treated with I3LA for 7 days.

## 3. Materials and Methods

### 3.1. Cell Culture

The human-derived primary cells were purchased from Bio Solution Co., Ltd. The human dermal fibroblasts (HFBs) and human epidermal keratinocyte (HKCs) were cultured in a humidified incubator at 37 °C and 5% CO_2_. The HFBs were cultured in FGM-2 fibroblast growth medium (Lonza, Basel, Switzerland), and the HKCs were cultured in KGM-Gold™ keratinocyte growth medium (Lonza, Basel, Switzerland). All of the cells were passaged 4–6 times.

### 3.2. Surface Functionalization Using Sulfo-SANPAH

Sulfo-SANPAH is a heterobifunctional photoreactive long-arm primary amine-nitrophenylazide crosslinker. The Sulfo-SANPAH was dissolved in deionized water at a working concentration of 10 mM. The polydimethylsiloxane (PDMS) surface was completely covered with the Sulfo-SANPAH solution and exposed to UV light at a light output of 30 mW/cm. The Sulfo-SANPAH was cross-linked at the double bond on the PDMS surface via the nitrophenylazide group during the UV treatment. After the UV exposure, the PDMS chambers were washed with phosphate-buffered saline (PBS). The fibronectin solution was then incubated at 4 °C for ≥3 h for further cross-linking. At the end of this process, the surfaces treated with the crosslinker were considered functionalized. A collagen solution containing cells was added to the PDMS chamber immediately after the surface functionalization was performed. When the collagen solution contacted the gels with the Sulfo-SANPAH-treated surfaces, the collagen fibers were crosslinked to the PDMS surface via the open NHS esters.

### 3.3. Soft Lithography

The skin-on-a-chip model manufacturing process was based on the soft lithography of PDMS. The preparations were performed as previously reported [43]. The PDMS that as used was prepared by mixing SYLGARD 184A and B (K1 solution, Gwangmyeong, Korea) at a 10:1 ratio. A porous membrane with a pore size of 0.48 μm (pore density: 3.5 × 10^6^ pores/cm^2^, polyester membrane, It4ip, Wallonia, Belgium) was laminated between the two PDMS layers (Figure 6d). The bottom of the two PDMS layers contained microfluidic channels that were designed to allow fluid flow by the process of gravity. An SU-8 master for imaging the microfluidic channel was fabricated as previously reported using photolithographic techniques (Figure 6a–c) [44].

### 3.4. Gravity Flow System for Pumpless Microfluidic Chip

The gravity flow system could control the angle and time (Figure 6e). The principle of operation is that the medium flows through the microfluidic channel of the pumpless microfluidic chip along the slope of the gravity flow system device and is supplied to the 3D culture scaffold via the membrane at the bottom of the culture chamber. As shown in the schematic diagram of Figure 6f, the system is connected to a PC that controls the motors and a chip holder that operates the repetitive fluctuation.

### 3.5. MTT Assay

The HKCs (1–5 × 10^6^ cells/mL) were grown in 96-well plates. The cells were treated with I3LA at various concentrations for 24 h. The medium was carefully aspirated. Then, 50 μL serum-free medium and 50 μL MTT reagent (ab211091, Abcam, Cambridge, UK) were added to each well. Because of the possible influence of phenol red, background control wells were prepared and treated with MTT reagent without cells. The plates were then incubated at 37 °C for 3 h. After the incubation, 150 μL MTT solvent was added to each well, and the plate was wrapped in foil and shaken on an orbital shaker for 15 min. The absorbance was measured at 590 nm. The corrected absorbance was calculated by subtracting the background of the culture medium from the readings. The cell cytotoxicity was calculated as a percentage of the *control*, as shown in Equation (1).
(1)$$\%\;\mathrm{Cytotoxicity}=\frac{\left(100\times \left(Control-Sample\right)\right)}{Control}$$

### 3.6. Fabrication of AD-HSEs

The AD-HSEs were generated using the method for skin fabrication that we previously reported on [4]. Rat tail collagen type I (Corning, New York, NY, USA), 10× DMEM medium, 0.5 N NaOH, HFB suspension (final cell concentration 5.0 × 10^5^ cell/mL), and media were mixed to neutralize the gel. To fabricate the dermis layer, the collagen–HFB suspension was seeded on the chip to a height of 3 mm. Thereafter, the dermal layer (DL) was cultured in FGM-2 fibroblast growth medium (Lonza, Basel, Switzerland) for 5 days, and the medium was changed every day. The HKCs were then seeded on the dermis layer (1.0 × 10^6^–5.0 × 10^6^ cell/cm^2^) and co-cultured for 2 days. KGM-Gold^TM^ Keratinocyte Growth Medium (Lonza) was supplied only above the DL-KCs, and FMG-2 media was supplied to the channel of the chip. We supplied the E-media to induce KC differentiation for between 7 and 14 days, and simultaneously, we provided an environment that was like that of real skin by exposing it to air. The E-media formulation included the following: DMEM/Ham’s F12, EGF-1 10 ng/mL, hydrocortisone 0.4 μg/mL, insulin 5 μg/mL, transferrin 5 μg/mL, 3,3,5-triiodo-L-thyonine sodium salt 2 × 10^−11^ M, cholera toxin 10^−10^ M, 10% (*v/v*) FBS, and 1% penicillin/streptomycin. The recombinant human IL-4 Protein and recombinant human IL-13 Protein (R&D system) treatment (15 ng/mL for both) was performed at the air exposure stage. All of the cultures were incubated in a 37 °C incubator with 5% CO_2_. A stock solution of I3LA was prepared by dissolving DMSO in solvent, and then, it was diluted with a 1.25 mM working solution and processed with medium.

### 3.7. H&E and IHC Staining

The HSE was fixed in formaldehyde at room temperature and embedded in paraffin. The paraffin-embedded tissue was sectioned with a microtome to a thickness of 4 μm, and the sectioned tissues were mounted on glass slides. Hematoxylin and eosin (H&E) or IHC staining was performed. For the H&E staining, the sections were stained with hematoxylin (Sigma-Aldrich Co., St. Louis, MO, USA) and eosin solutions (Sigma-Aldrich Co., St. Louis, MO, USA). The stained tissues were observed using an inverted optical microscope (Olympus, IX73-F22PH, Tokyo, Japan). For the IHC staining, after replacing paraffin with xylene, the tissues were treated with 100%, 90%, 80%, 60%, and 50% ethanol for between 3 and 5 min for each step, and finally, they were immersed in DW for between 3 and 5 min for hydration.

The tissues were then blocked with 3% bovine serum albumin (BSA) and treated with primary antibodies. Antibodies against filaggrin (ab81468), loricrin (ab85679), and involucrin (ab53112) were used. The secondary antibody that was used was goat anti-rabbit IgG H&L (HRP) (ab205718) at a 1:200 dilution. The sections were incubated with a horseradish peroxidase (HRP)-conjugated secondary antibody and the target proteins were visualized using 3,3-diaminobenzidine (DAB). The sections were stained with hematoxylin solution for nuclear staining. The stained slides were photographed using an optical microscope (Olympus, IX73-F22PH, Tokyo, Japan) equipped with an optical camera (Olympus, DP73-ST-SET). Five random shots of each sample were subjected to an IHC image analysis using the free ImageJ Fiji software. The measurements were analyzed using the PRISM program.

### 3.8. Measurement of HSE Contraction

The formation of the 3D-cultured HSEs seeded in the culture chamber of a pumpless skin-on-a-chip model was photographed using a camera to measure the degree of contraction of the entire tissue area. The cultured tissue was observed daily, and the area was measured using the ImageJ Fiji program. Six samples were measured, and the difference in contraction for each sample was calculated by calculating the mean value, standard deviation, and *p*-value of the measurements using PRISM. The data are visualized as graphs.

### 3.9. Quantitative Real-Time PCR (qPCR)

This procedure was performed as previously reported [4]. Next, 1 mL TRIzol™ Reagent (Invitrogen, Waltham, MA, USA) was added to the cultured HSE tissue. The quality and quantity of RNA were assessed using a Nanodrop spectrophotometer (SpectraMax M2, Molecular Devices, LLC., San Jose, CA, USA). The mRNA was synthesized into cDNA via RT-PCR using an amfiRivert cDNA Synthesis Platinum Master Mix (GenDEPOT, Katy, TX, USA). The synthesized cDNA was quantitatively analyzed with real-time PCR using LightCycler 480 SYBR Green I Master (Roche, Basel, Switzerland) and a LightCycler 480 Instrument II (Roche, Basel, Switzerland). The GAPDH was used as the housekeeping gene. The relative expression values were calculated using the delta-delta CT method. The primers that were used for the qPCR are listed in Table 2.

### 3.10. Scanning Electron Miscroscopy Images

The samples were fixed overnight with 2.5% glutaraldehyde, which was followed by 1% osmium tetroxide. A series of ethanol concentrations (50, 70, 80, 90, 95, and 100% for 15 min each) were used for the dehydration. The samples were maintained in hexamethyldisilazane (HMDS) (Sigma-Aldrich, St. Louis, MO, USA) for 10 min at 15–25 °C. The samples were then dried in a fume hood and sputter-coated with Au before being imaged using an SEM (JEOL Ltd., Tokyo, Japan).

### 3.11. Statistical Analysis

The data from at least three independent experiments are expressed as mean ± standard deviation. The statistical analysis was performed using PRISM (version 9.3.0, GraphPad Software Inc., Santa Clara, CA, USA). The statistical significance was determined using one-way and two-way ANOVAs. The MTT assay and DAB intensity results were using determined using a one-way ANOVA, which was followed by Dunnett’s and Bonferroni’s comparisons tests. The qPCR results were determined using a two-way ANOVA and Dunnett’s comparisons test. *p*-values of <0.05 were considered to be statistically significant (* *p* < 0.05; ** *p* < 0.01; *** *p* < 0.001; **** *p* < 0.0001).

## 4. Conclusions

Topical steroids commonly cause side effects, but they remain the representative anti-inflammatory agents for AD. Novel treatment strategies using the skin microbiota may be safer and more effective. As part of the basic research to utilize the skin microbiota to develop topical or systemic drugs from skin microbial metabolites and to produce useful therapeutics, I3LA was investigated for its efficacy as an AD therapeutic agent. The I3LA treatment for 7 days had a positive effect on AD and the allergy-related gene biomarkers, but it did not affect the skin barrier-related mRNA expression. It did result in an increase in the expression of the major constituent proteins of the skin barrier. However, the I3LA treatment for more than 14 days resulted in surface roughness and keratin shedding, and it did not affect the AD-related mRNA or the skin barrier protein expression. Therefore, the short-term use of I3LA for approximately one week is considered to be effective in suppressing AD.

In addition, we determined that I3LA inhibited the contraction of HSEs. The significant anti-contractile effect of I3LA was attenuated by the addition of cytokines. The contractile inhibitory effect of I3LA was reduced by the cytokines in the following order of magnitude: IL-4 and IL-13 stimulation, the IL-13 only treatment, and the IL-4 only treatment. In this study, it was not possible to clarify the mechanism by which I3LA, a microbial metabolite, was involved in the contraction HSEs suppression and how it affected the expression of the AD-related mRNAs such as *TSLP* and *IL-6*. In the future, extensive research should be conducted to understand the biological effects of these metabolites on contraction, and to understand the current and potential treatment approaches and potential value of skin microbial metabolites in regulating such conditions.

## Figures and Tables

**Figure 1 ijms-23-13520-f001:**
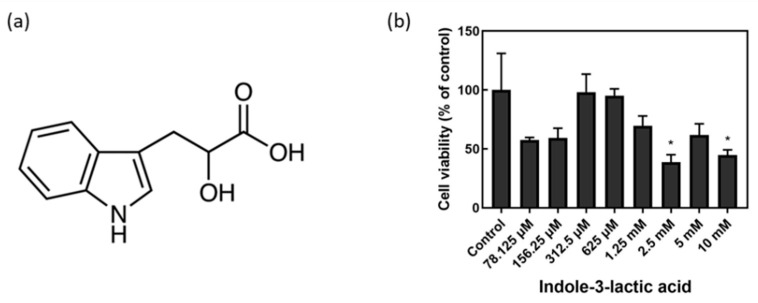
Indole-3-lactic acid (I3LA) structure and MTT assay results. (**a**) Formula for I3LA (molecular weight is 205.21 g/M; empirical formula (hill notation: C11H11NO3). (**b**) Calculated concentration versus cell viability data for I3LA toxicity in keratinocytes using MTT assay (50% or more cell viability was assumed to be safe). (*n* = 3; * *p* < 0.05; control group vs. several experimental groups).

**Figure 2 ijms-23-13520-f002:**
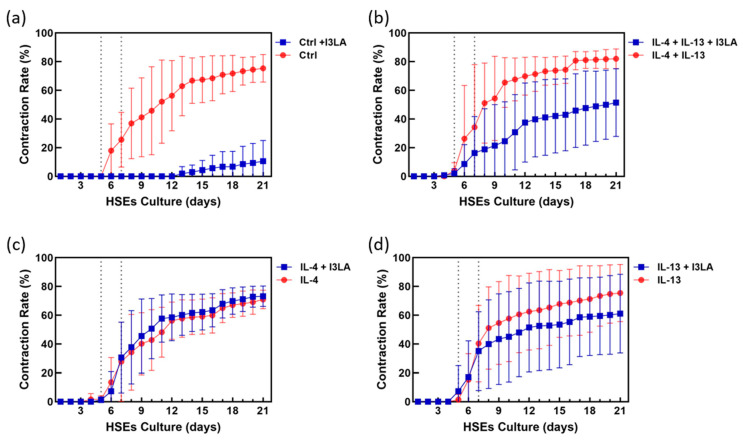
Contraction rate changes during incubation between group that was treated with I3LA (■) and group that was treated without I3LA (●). (**a**) Control conditions; (**b**) IL-4 and IL-13 treatment; (**c**) IL-4 only treatment; (**d**) IL-13 only treatment.

**Figure 3 ijms-23-13520-f003:**
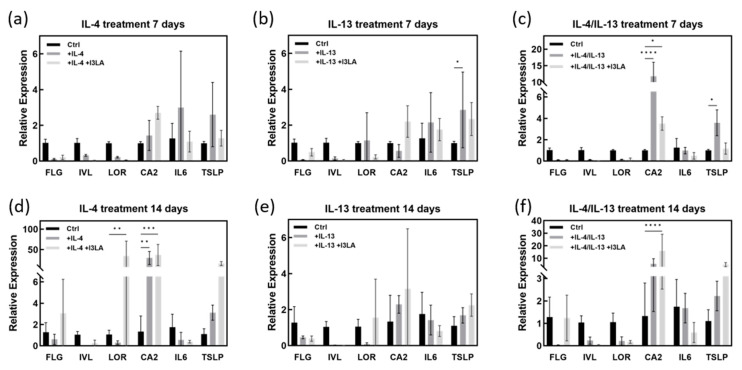
Processed real-time PCR results of I3LA treatment in AD-HSEs stimulated with IL-4 and/or IL-13. Gene expression results for AD-HSEs treated with IL-4 and IL-13 at different concentrations and for different time periods, the control group, and the group treated with I3LA. (**a**) Group treated with IL-4 for 7 days, (**b**) group treated with IL-13 alone for 7 days, (**c**) group treated with combined IL-4 and IL-13 for 7 days, (**d**) group treated with IL-4 alone for 14 days, and (**e**) group treated with IL-13 alone for 14 days. (**f**) FLG, IVL, LOR, CA2, IL-6, and TSLP gene expression comparison graphs in the group treated with combined IL-4 and IL-13 for 14 days (*n* = 3; two-way ANOVA; Dunnett’s comparisons t test; * *p* < 0.05; ** *p* < 0.01; *** *p* < 0.001; **** *p* < 0.0001).

**Figure 4 ijms-23-13520-f004:**
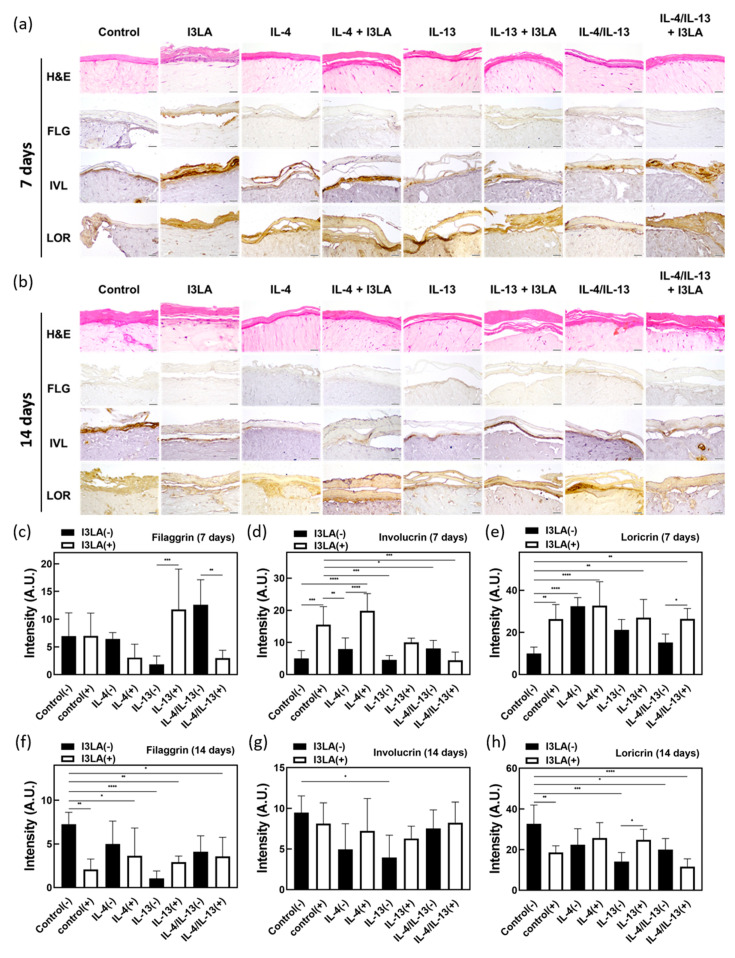
Images of H&E and IHC staining results for AD-HSEs stimulated with various concentrations of IL-4 and IL-13 and cultured for 7 and 14 days, depending on the presence or absence of I3LA treatment, and graphs quantifying IHC DAB color development. (**a**) IHC staining images of epidermal morphogenetic proteins in AD-HSEs stimulated with IL-4 and IL-13 for 7 days and which were treated with I3LA. (**b**) IHC staining of epidermal morphogenetic proteins in AD-HSEs stimulated with IL-4 and IL-13 for 14 days and which were treated with I3LA (scale bar size in (**a**,**b**) = 50 μm). (**c**–**h**) A comparison graph was obtained by quantifying the results of filaggrin, involucrin, and loricrin IHC, which are epidermal barrier proteins, by filtering only DAB using ImageJ software. (**c**–**e**) Results when AD-HSEs were cultured for 7 days. (**f**–**h**) Results when AD-HSEs were cultured for 14 days. (*n* = 5; one-way ANOVA; Dunnett’s and bonferroni’s comparisons test; * *p* < 0.05; ** *p* < 0.01; *** *p* < 0.001; **** *p* < 0.0001).

**Figure 5 ijms-23-13520-f005:**
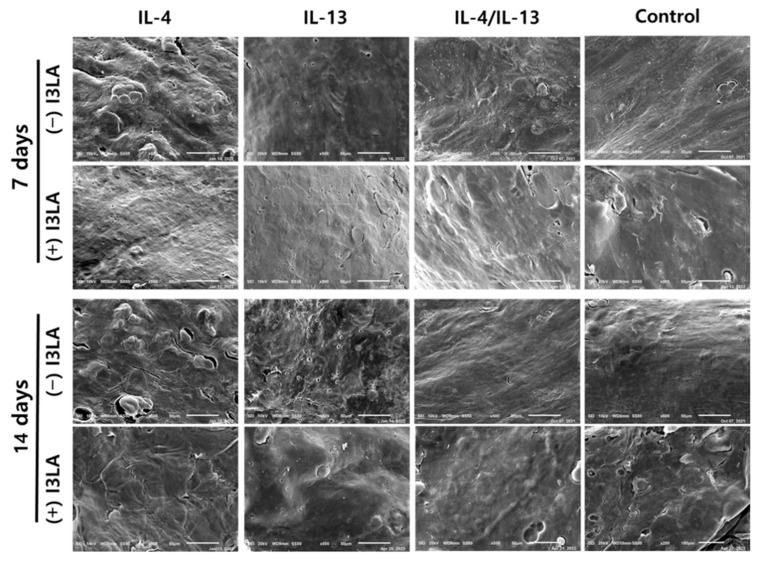
AD-HSEs were cultured under different conditions, and the surface was photographed using scanning electron microscopy (SEM). SEM results were taken from air-exposed epidermal surface views. Scale bar represents 50 μm.

**Figure 6 ijms-23-13520-f006:**
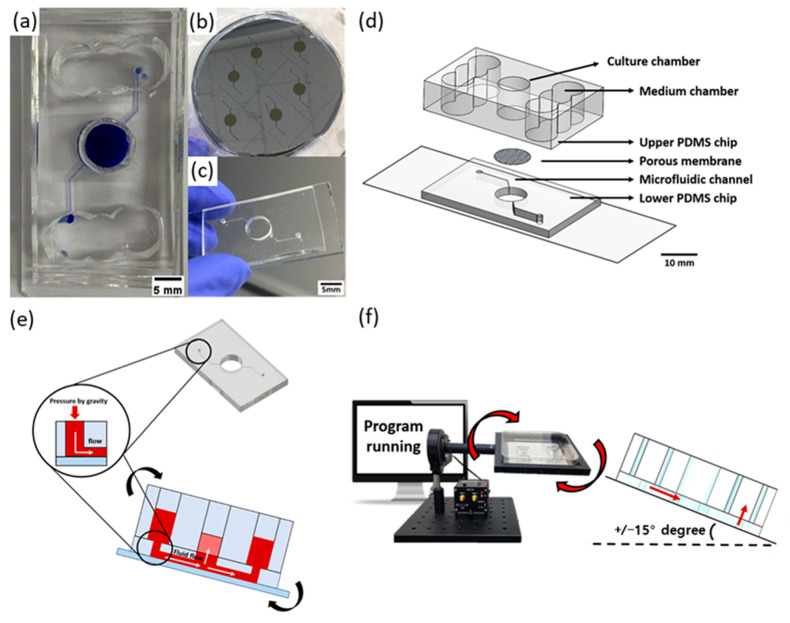
A schematic diagram of a microfluidic skin-on-a-chip model before assembly and a gravity flow system. (**a**) Upper PDMS chip. (**b**) Microfluidic channels patterned mask mold. (**c**) The lower chip with the microfluidic channel pattern. (**d**) Configuration diagram of pumpless microfluidic skin-on-a-chip model. (**e**) A schematic diagram of a microfluidic skin-on-a-chip model before its assembly. (**f**) Views of the front and side of the gravity flow system. It works by shaking both sides at 15° degrees. A 15° degree tilt causes the medium to circulate through the microfluidic channel [4,43].

**Table 1 ijms-23-13520-t001:** Differences in contraction compared to control (%).

Treated Cytokine Conditions	(−) I3LA	(+) I3LA
Control *	0.00	−71.27
IL-4	−10.88	−8.64
IL-13	−6.38	−20.75
IL-4 & IL-13	0.14	−30.37

* The reference value is the contraction in the control (81.87%).

**Table 2 ijms-23-13520-t002:** List of sequence of primers for Real-time quantitative PCR.

Gene	Forward Primer	Reverse Primer
*GAPDH*	5′-CTCCTCTGACTTCAACAGCG-3′	5′-GCCAAATTCGTTGTCATACCAG-3′
*FLG*	5′-GGAGTCACGTGGCAGTCCTCACA-3′	5′-GGTGTCTAAACCCGGATTCACC-3′
*IVL*	5′-CCGCAAATGAAACAGCCAACTCC-3′	5′-GGATTCCTCATGCTGTTCCCAG-3′
*LOR*	5′-GTCTGCGGAGGTGGTTCCTCT-3′	5′-TGCTGGGTCTGGTGGCAGATC-3′
*CA2*	5′-AACAATGGTCATGCTTTCAACG-3′	5′-TGTCCATCAAGTGAACCCCAG-3′
*IL-6*	5′-AGACAGCCACTCACCTCTTCAG-3′	5′-TTCTGCCAGTGCCTCTTTGCTG-3′
*TSLP*	5′-AGTGGGACCAAAAGTACCGAGTT-3′	5′-GGATTGAAGGTTAGGCTCTGG-3′

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
