# Peer review of "Effects of Indole-3-Lactic Acid, a Metabolite of Tryptophan, on IL-4 and IL-13-Induced Human Skin-Equivalent Atopic Dermatitis Models"

_ijms, 2022, doi:10.3390/ijms232113520_

Round 1

Reviewer 1 Report

It is a complex and interesting paper. I would you clarify what are the long term side effects of dupilumab that you mentioned at line 59.

Author Response

â–¶ Thanks for your helpful comment. Dufilumab is very effective, but side effects have been reported in some patients. Dufilumab is treated with topical corticosteroids, which have side effects when treated long-term, and side effects for long-term use are also mentioned generically. A brief description of what was reported in this regard is as follows:
Commonly observed abnormal reactions include local injection site reactions, conjunctivitis, headache, nasopharyngitis.1

Very rarely reported side effects include alopecia areata and cicatricial extropion.2,3

A recently published case report also described a case of moderate-to-severe AD in which a facial rash developed after dupilumab treatment, which was not previously observed in the literature.4

Side effects have also been reported with long-term corticosteroid use.5

There have been many reported side effects of short-term/long-term treatment with dufilumab. The text briefly mentions the side effects and adds references to the literature in which the side effects have been reported. We corrected the contents as follows.

However, since side effects such as local injection site reactions, conjunctivitis, headache, nasopharyngitis, alopecia areata, and cicatricial extropion may be prolonged during use [15-18], substances and treatments that can be used for a long time without toxicity are required.”

1 Lee, Ji Hyun, Sag Wook Son, and Sang Hyun Cho. "A comprehensive review of the treatment of atopic eczema." Allergy, asthma & immunology research 8.3 (2016): 181-190.

2 Mitchell, Krystal, and Jacob Levitt. "Alopecia areata after dupilumab for atopic dermatitis." JAAD case reports 4.2 (2018): 143-144.

3 Barnes, Alexander C., Alexander D. Blandford, and Julian D. Perry. "Cicatricial ectropion in a patient treated with dupilumab." American Journal of Ophthalmology Case Reports 7 (2017): 120-122.

4 Krathen, Richard A., and Sylvia Hsu. "Failure of omalizumab for treatment of severe adult atopic dermatitis." Journal of the American Academy of Dermatology 53.2 (2005): 338-340.

5 Yasir, Muhammad, et al. "Corticosteroid adverse effects." (2018).

Reviewer 2 Report

This study investigated the effects of indole-3-lactic acid (I3LA) treatment on atopic dermatitis (AD) using human skin equivalent models. The study is well performed. I have some comments as follows:

1. If possible, please reconsider the title to emphasize the importance of I3LA treatment.

2. Page 1: Please move T helper 2 from Line 33 to Line 31 to explain Th2.

3. Page 2, Line 70: Please capitalize and italicize “staphylococcus aureus”.

4. Page 3, Line 101-102: How to assume that viabilities of higher than 50% were safe? (Please cite the reference for your assumption).

5. Figure legend 1b: Please explain the symbol *.

6. Figure 4: Are there any statistical differences? If yes, please denote them.

7. Page 9, Line 288: Please indicate the time interval for I3LA treatment for MTT assay.

8. Page 11, Line 366: Why did the authors use the two-way ANOVA test rather than one-way ANOVA? Additionally, please indicate the post-hoc test.
